# Dynamics of Antibiotic Resistance of *Streptococcus pneumoniae* in France: A Pediatric Prospective Nasopharyngeal Carriage Study from 2001 to 2022

**DOI:** 10.3390/antibiotics12061020

**Published:** 2023-06-06

**Authors:** Alexis Rybak, Corinne Levy, Naïm Ouldali, Stéphane Bonacorsi, Stéphane Béchet, Jean-François Delobbe, Christophe Batard, Isabelle Donikian, Marie Goldrey, Jessica Assouline, Robert Cohen, Emmanuelle Varon

**Affiliations:** 1Activ, Association Clinique et Thérapeutique Infantile du Val-de-Marne, 94000 Créteil, France; corinne.levy@activ-france.fr (C.L.); naim.ouldali@aphp.fr (N.O.); christophe.batard@activ-france.fr (C.B.); robert.cohen@activ-france.fr (R.C.); 2Afpa, Association Française de Pédiatrie Ambulatoire, 45000 Orléans, France; delobbejf@yahoo.fr (J.-F.D.); isabelledonikian@free.fr (I.D.); dr.m.goldrey@orange.fr (M.G.); jess.assouline@gmail.com (J.A.); 3Pediatric Emergency Department, Trousseau Hospital, Sorbonne Université, Assistance Publique–Hôpitaux de Paris, 75012 Paris, France; 4Clinical Epidemiology Unit, Eceve Inserm UMR-S 1123, Robert Debré University Hospital, Université de Paris, Assistance Publique–Hôpitaux de Paris, 75010 Paris, France; 5IMRB-GRC GEMINI, Institut Mondor de Recherche Biomédicale-Groupe de Recherche Clinique Groupe d’Etude des Maladie Infectieuses Néonatales et Infantiles, Université Paris Est, 94000 Créteil, France; emmanuelle.varon@chicreteil.fr; 6GPIP, Groupe de Pathologie Infectieuse Pédiatrique, 06200 Nice, France; 7CRC, Clinical Research Center, Centre Hospitalier Intercommunal de Créteil, 94000 Créteil, France; 8Department of General Pediatrics, Pediatric Infectious Disease and Internal Medicine, Robert Debré University Hospital, Université de Paris, Assistance Publique–Hôpitaux de Paris, 75019 Paris, France; 9IAME, Infection, Antimicrobials, Modelling, Evolution, Inserm UMR 1137, Paris University, 75018 Paris, France; 10Microbiology Unit, Robert-Debré University Hospital, Assistance Publique-Hôpitaux de Paris, Université de Paris, 75019 Paris, France; stephane.bonacorsi@aphp.fr; 11Laboratory of Medical Biology and National Reference Centre for Pneumococci, Intercommunal Hospital of Créteil, 94000 Créteil, France

**Keywords:** pneumococcal nasopharyngeal carriage, children, acute otitis media, PCV impact, third-generation pneumococcal conjugate vaccine, next-generation pneumococcal conjugate vaccine

## Abstract

Epidemiological surveillance of nasopharyngeal pneumococcal carriage is important for monitoring serotype distribution and antibiotic resistance, particularly before and after the implementation of pneumococcal conjugate vaccines (PCVs). With a prospective surveillance study in France, we aimed to analyze the dynamics of pneumococcal carriage, antibiotic susceptibility and serotype distribution in children aged 6 to 24 months who had acute otitis media between 2001 and 2022 with a focus on the late PCV13 period from May 2014 to July 2022. Trends were analyzed with segmented linear regression with autoregressive error. For the 17,136 children enrolled, overall pneumococcal carriage was stable during the study. During the late PCV13 period, the five most frequent serotypes were all non-PCV13 serotypes: 15B/C (14.3%), 23B (11.0%), 11A (9.6%), 15A (7.4%) and 35B (6.5%). During the same period, we observed a rebound of penicillin non-susceptibility (+0.15% per month, 95% confidence interval, +0.08 to 0.22, *p* < 0.001). Five serotypes accounted for 64.4% of the penicillin non-susceptible strains: 11A (17.5%), 35B (14.9%), 15A (13.9%), 15B/C (9.9%) and 19F (8.2%); non-PCV13/PCV15 accounted for <1%, and non-PCV15/PCV20 accounted for 28%. The next generation PCVs, particularly PCV20, may disrupt nasopharyngeal carriage and contribute to decreasing the rate of antibiotic resistance among pneumococci.

## 1. Introduction

The nasopharynx, particularly in children, is the ecological niche of pneumococci and the site of selective pressure due to antibiotic exposure and conjugate vaccines [1]. Nasopharyngeal (NP) carriage is a prerequisite for pneumococcal disease [1,2]. Infections due to *S. pneumoniae* include acute otitis media (AOM) [3], which is in many countries the main cause for antibiotic prescription in children [4]. Pneumococcus is also implicated in other less frequent respiratory tract infections, such as sinusitis and pneumonia, and invasive pneumococcal infections. The monitoring of *S. pneumoniae* strains involved in non-invasive infections is a challenge. Indeed, tympanocentesis is not performed in most countries for first-line treatment of AOM [5]. Therefore, carriage surveillance has been proposed as a proxy to monitor the serotype distribution and antibiotic resistance of pneumococci for AOM [5], especially after pneumococcal conjugate vaccine implementation [6]. In many countries, antibiotics are recommended for AOM in young children. The resistance of pneumococcus and *Haemophilus influenzae* to antibiotics is among the main determinants for the choice and the dosage of first-line antibiotics. Furthermore, NP carriage studies are important to try to understand the complex interactions of PCV implementation and antibiotic use in *S. pneumoniae* carriage and infections.

After the implementation of pneumococcal conjugate vaccines (PCVs), PCV7 and then PCV13, major changes to pneumococcal diseases and carriage occurred. First, a decrease in invasive and non-invasive pneumococcal infections such as AOM was reported [7] along with a stable or slight decrease in pneumococcal NP carriage [8,9]. Concurrently, the prevalence of PCV serotypes, frequently penicillin non-susceptible, greatly decreased. Over time, a serotype replacement occurred [9].

Despite the establishment of several national programs for a more judicious and appropriate use of antibiotics [10], the French population still ranks among the main antibiotic consumers in Europe [10,11]. Antibiotic use increases the risk of antibiotic non-susceptible *S. pneumoniae* carriage and infections [12].

Therefore, with this ongoing NP carriage study performed in France from 2001 to 2022, we report the dynamics of pneumococcal NP carriage and antibiotic resistance before third-generation PCV implementation in the COVID-19 pandemic era [13].

## 2. Results

### 2.1. Patient Characteristics

Between September 2001 and July 2022, swab samples were obtained from 17,136 children with AOM (patient characteristics are in Table 1). The median age was 12.8 months (interquartile range 9.3–17.3), and 9159 were boys (53.4%). During the study period, several statistically significant changes occurred between the pre-PCV7 period and the late PCV13 period (Table 1). The proportion of children attending daycare increased from 33.5% to 59.1% (+0.16% per month over the study period, 95% CI +0.14 to +0.19, *p* < 0.001), whereas the proportion of children who received antibiotics in the three months before enrollment decreased from 47.2% to 36.3% (−0.06% per month over the study period, 95% CI −0.10 to −0.02, *p* = 0.001). Of note, the use of broad-spectrum antibiotics decreased during the study period (Appendix A).

### 2.2. Carriage and Antibiotic Resistance of S. pneumoniae

The overall pneumococcal carriage rate (mean 57.4%, 9839/17,136) remained stable over the study period (Table 2, Figure 1). However, the penicillin non-susceptibility rate among the *S. pneumoniae* isolated strains was greatly modified (Figure 2 and Table 2). Although antibiotic resistance decreased during the PCV7 period and during the early PCV13 period, after May 2014, the penicillin non-susceptibility rate increased (+0.15% per month, 95% CI +0.08 to +0.22, *p* < 0.001, Figure 2). We found similar trends for erythromycin non-susceptibility during these periods.

### 2.3. Serotype Distribution and Relation with Penicillin Non-Susceptibility

After the implementation of PCVs, despite a marked decrease in PCV7 serotypes (from 44.6% to 2.0%) and PCV13 serotypes (from 37.9% to 4.5%), they were still isolated in the late PCV13 period (Table 2). Three vaccine serotypes accounted for 6.5% of the pneumococcus cases in the late PCV13 period: 19F (3.3%), 19A (2.0%) and 3 (1.2%).

The distribution of the most frequently isolated *S. pneumoniae* serotypes by period varied greatly. In the late PCV13 period, the five most frequent serotypes were all non-PCV13 (Table 2, Figure 3, Appendix A) and accounted for 48.8% of the strains: 15B/C (14.3%), 23B (11.0%), 11A (9.6%), 15A (7.4%) and 35B (6.5%).

Changes in the serotype distribution among the penicillin non-susceptible *S. pneumoniae* (PNSP) strains by period were similar (Table 2 and Figure 4). In the late PCV13 period, five serotypes accounted for 64.4% of the PNSP strains: 11A (17.5%), 35B (14.9%), 15A (13.9%), 15B/C (9.9%) and 19F (8.2%); non-PCV13/PCV15 accounted for <1%, and non-PCV15/PCV20 accounted for 28%. During the same period, the main erythromycin non-susceptible serotypes were 15A (22.5%), 15B/C (11.4%), 19F (11.2%), 24F (8.0%) and 23A (7.7%). Of note, non-typable serotypes with a penicillin non-susceptibility rate of 74.5% accounted for 8.3% of the erythromycin non-susceptible strains.

### 2.4. Dynamics of Predominant Serotypes during the Late PCV13 Period: Carriage and Penicillin Non-Susceptibility

We identified three predominant serotypes during the late PCV13 period: 15B/C, 23B and 11A; each one had its own dynamics of progression and profile of resistance. An increase in carriage was observed for serotypes 15B/C and 23B, while the proportion of PNSP increased for serotype 23B and especially for serotype 11A (Appendix A).

### 2.5. Risk Factors of Antibiotic Non-Susceptible S. pneumoniae

During the late PCV13 period, PNSP carriage was associated with recent use of antibiotics (aOR 1.56, 95% CI 1.28 to 1.88, *p* < 0.001), daycare center (aOR 1.25, 95% CI 1.07 to 1.47, *p* = 0.006) and history of AOM (aOR 1.36, 95% CI 1.13 to 1.64, *p* = 0.001, Appendix A). Erythromycin non-susceptible *S. pneumoniae* carriage was associated with recent use of antibiotics (aOR 1.36, 95% CI, 1.10 to 1.68, *p* = 0.004) and history of AOM (aOR 1.27, 1.03 to 1.56, *p* = 0.02). Age <1 year was not associated with carriage of antibiotic non-susceptible strains.

### 2.6. Impact of Non-Pharmaceutical Interventions for COVID-19

We introduced a period from April 2020 after the first national lockdown for COVID-19 and observed no significant change in overall pneumococcal carriage, rate of penicillin non-susceptibility or rate of erythromycin non-susceptibility among *S. pneumoniae* strains (Appendix A).

## 3. Discussion

With this 21-year extended prospective study before and during PCV implementation, we provide data on the dynamics of pneumococcal carriage in children as well as resistance to antibiotics from 2001 to 2022. Pneumococcal carriage, serotype distribution and antibiotic resistance were affected at different levels by the complex interactions of PCVs, antibiotic consumption and daycare attendance modalities. In our study, several results should be highlighted to help unravel these interactions.

First, we observed a striking decrease in pneumococcal resistance when PCV7 and then PCV13 were implemented, as in other studies [14,15]. This finding was easily explained by the following: a total of 75% of pneumococcal penicillin non-susceptibility was due to PCV7 serotypes before PCV7 implementation (mainly serotypes 19F [18.3%], 6B [17.9%], 23F [17.9%] and 14 [15.6%]), and 50% to PCV13 serotypes before PCV13 implementation (mainly serotype 19A [43.9%]). The effectiveness of the PCVs against pneumococcal infections but also carriage has resulted in most disappearing and has considerably decreased the frequency of carriage of 19F, 19A and 3, which remained at a low level despite high vaccine coverage [16].

Furthermore, this decrease in pneumococcal resistance was facilitated by a decrease in antibiotic consumption. However, the population in France is among the major community users of antibiotics (4th among 28 European countries in 2021) [11]. The differences between the European countries could explain the lower rate of antibiotic resistance observed in Belgium: only 17.7% of *S. pneumoniae* strains isolated from healthy children attending daycare between 2016 and 2018 were PNSP [17]. Of note, in our study, the decrease in antibiotic resistance occurred despite a significant increase in daycare attendance, which increased from 33.5% to 59.1% during the study period. This increase agrees with the national data that shows daycare attendance doubled in children aged <3 years between 2002 and 2021 (from 9% to 18%) [18]. These children have much higher odds of AOM, which explains the overrepresentation of this population in our study [19]. Crowding, poor hygiene practices, high person-to-person contact, viral infections and antibiotic use may promote the selection and spread of antibiotic-resistant *S. pneumoniae* strains in daycare centers [20,21].

Second, since May 2014, four to five years after PCV13 implementation, unfortunately, antibiotic resistance regularly increased without an increase in antibiotic consumption. This observation was mainly due to the emergence of non-PCV13 serotypes among which antibiotic non-susceptible strains are frequently found. Indeed, during the late PCV13 period, penicillin non-susceptibility involved mainly serotypes 11A (17.5%), 35B (14.9%), 15A (13.9%), 15B/C (9.9%) and 19F (8.2%). The new PCVs, if they are active on carriage of the added serotypes, could contribute to decreasing penicillin non-susceptibility: minimal for PCV15 (serotypes 22F and 33F are rarely carried and rarely resistant to penicillin) but more pronounced for PCV20 (serotypes 15BC and 11A are among the most carried and often antibiotic resistant).

We observed no significant change in *S. pneumoniae* carriage after non-pharmaceutical intervention implementations for COVID-19 despite a major decrease in invasive pneumococcal disease (IPD) incidence observed after non-pharmaceutical intervention implementations [22]. Our results agree with several studies that linked the decrease in IPD incidence to a reduction in respiratory viral infections rather than a modification of pneumococcal carriage [23,24,25]. Furthermore, we observed no significant change in pneumococcal antibiotic resistance during the COVID-19 period from April 2020 to July 2022 compared to the trend observed from May 2014 to March 2020. This result contrasts with a surveillance study in Spain where an increase in antibiotic resistance was observed in 2022 in pneumococcal strains isolated from infections in adults [26]. The authors hypothesized that the generic use of antibiotics to prevent bacterial co-infections in adults with COVID-19 might explain these results. By contrast, antibiotic use in France decreased in 2020, especially in children aged <5 years, and increased to a pre-pandemic level in 2021 [27].

Several limitations need to be discussed. In our work, we studied antibiotic susceptibilty in pneumococcal carriage among children with AOM rather than using isolates from tympanocentesis or IPD. We are aware that all pneumococcal serotypes do not have the same disease potential [28], and therefore, serotype distribution in carriage studies is different from that in IPD studies. Among the serotypes implicated in pediatric IPD in France in 2021, the most frequent were the non-PCV13 serotypes 24F, 10A, 23B, 15B/C, 15A and 12F, while PCV13 serotypes 3, 19F and 19A continued to be isolated [29]. Trends in serotype replacement and antibiotic resistance in carriage are similar to those in IPD. As discussed previously, France has high PCV13 vaccine coverage, major community antibiotic consumption and a high proportion of children attending daycare. Furthermore, clonal changes in the pneumococcal population in carriage may differ from country to country. This combination of factors may prevent the generalization of our results. Finally, whole-genome sequencing and multi-locus sequencing typing (MLST) are not routinely performed in our study. These techniques could strengthen the relation between serotype and antibiotic resistance. For example, GPSC10 was recently identified as a predominant lineage among serotype 24F strains isolated from carriage and IPD in France [30]. This lineage has spread to Europe and other continents and is frequently multidrug-resistant [30]. Similarly, an emergence of penicillin-resistant strains of serotype 11A, linked to the clone ST6521, has been described in Spain [26,31]. Further studies with a similar approach may provide additional information to understand the major increase in penicillin non-susceptibility of serotype 11A or the spread of serotype 15B/C in France.

## 4. Material and Methods

### 4.1. Study Design

Between September 2001 and July 2022, 161 pediatricians throughout France participated in this prospective population-based surveillance study. Pediatricians are part of a research and teaching network (ACTIV) [32]. The methodology was previously described [12,23,33]. From September to July of each subsequent year, we enrolled children from both sexes with AOM who were aged 6 to 24 months. The diagnostic criteria for AOM included the algorithm proposed by Paradise for acute suppurative otitis media (effusion plus marked redness, marked bulging or moderate redness and bulging) associated with fever and/or otalgia and/or irritability [34]. Previous studies have shown that children with AOM have higher rates of pneumococcal carriage compared to healthy children [35,36]. Enrollment of these children allowed us to more easily monitor the serotype distribution and antibiotic susceptibility over time by reducing the number of children swabbed. We excluded children with antibiotic treatment within 7 days before enrolment, severe underlying disease or inclusion in the study during the previous 12 months. The physician collected the medical history, antibiotic consumption during the 3 months before inclusion and type of antibiotics used, immunization history, daycare attendance and clinical examination findings on an electronic case report form.

### 4.2. Ethics

Written informed consent was obtained from the parents or guardians. The study was approved by the Saint-Germain-en-Laye Hospital Ethics Committee. This study was registered at ClinicalTrials.gov (NCT04460313).

### 4.3. Microbiological Investigations

The techniques of NP swabbing and microbiological methods were previously described [12,23,33] and remained unchanged over the whole study period. The specimens were placed in transport medium and transferred within 48 h for analysis at the National Reference Center for Pneumococci (NRCP; Centre Hospitalier Intercommunal de Créteil, Créteil, France) or at Robert Debré Hospital (Paris, France). *S. pneumoniae* strains were identified with morphology and standard methods. The susceptibility of the pneumococcal isolates to penicillin was determined with gradient concentration strips. The isolates were classified as penicillin-susceptible (minimal inhibitory concentration (MIC) ≤ 0.06 mg/L), increased exposure (MIC 0.12–2.0 mg/L) or penicillin-resistant (MIC > 2 mg/L) according to the European Committee on antimicrobial susceptibility testing [37]. Similarly, we defined erythromycin-susceptible strains (MIC ≤ 0.25 mg/L) and erythromycin-resistant strains (MIC > 0.25 mg/L) [37]. All pneumococcal strains were serotyped at the NRCP with latex agglutination and antiserum provided by the Statens Serum Institute (Copenhagen, Denmark).

### 4.4. Definitions

We defined PCV7 serotypes (4, 6B, 9V, 14, 18C, 19F and 23F), PCV13 + 6C serotypes (PCV7 serotypes with 1, 3, 5, 6A, 6C, 7F and 19A), PCV15 serotypes (PCV13 serotypes with 22F and 33F) and PCV20 serotypes (PCV15 serotypes with 8, 10A, 11A, 12F and 15B/C). Serotype 6C was included in the PCV13 serotypes because studies suggested immunological and clinical activity of the vaccine for this serotype [38,39]. Serotypes 15B and 15C were considered a single serotype (15B/C) because their capsule is quickly interchangeable [40].

We defined the following periods: The “pre-PCV7 period” from September 2001 to December 2002, the “targeted PCV7 period” from January 2003 to May 2006, the “PCV7 period” from June 2006 to May 2010, the “early PCV13 period” from June 2010 to April 2014 and the “late PCV13 period” from May 2014 to July 2022. We performed a second analysis with a “late PCV13 period” from May 2014 to March 2020” and a “COVID-19 period” from April 2020 to July 2022. Of note, PCV coverage was approximately 60% in 2005 (during the targeted PCV7 period), 80% in 2007 (during the PCV7 period) and >90% since 2011 (during the early and late PCV13 periods) [41].

### 4.5. Statistical Analyses

Data were entered by using 4D (v6.4 to v17.4) and analyzed with Stata SE v15.1 (Stata Corp., College Station, TX, USA).

Data were analyzed using segmented linear regression with autoregressive error [42,43]. We analyzed monthly data to provide optimal precision. The model took into account autocorrelation and the trend before and after PCV implementations, which were considered in the model by including dummy variables for each period. The post-intervention trend change was estimated by comparing observations with the expected trend if the intervention had not occurred. Furthermore, we analyzed the monthly carriage and penicillin non-susceptibility rates of the emerging serotypes from the late PCV13 period. For these non-vaccine serotypes, we identified knots (specific time points when a trend change occurred) to select the model providing the best fit. We used the same method to define the beginning of the late PCV13 period, which corresponds to a rebound in both penicillin and erythromycin non-susceptibility. Finally, we performed a second analysis of pneumococcal carriage and antibiotic non-susceptibility by including a trend change and an immediate change after the implementation of non-pharmaceutical interventions against COVID-19 [44].

We analyzed the factors associated to the carriage of antibiotic non-susceptible *S. pneumoniae*; clinical variables with a *p*-value < 0.20 on univariate analysis were included in the multinomial model, estimating the adjusted odds ratios (aORs) and 95% confidence intervals (CIs).

The tests were 2-sided. We considered *p*-values < 0.05 statistically significant.

## 5. Conclusions

After the implementation of PCV7 and PCV13, pneumococcal antibiotic resistance significantly decreased in France. However, several years after PCV13 implementation, antibiotic resistance has been increasing due to the emergence of resistant non-vaccine serotypes. Expected changes encompassing replacement together with uprising antibiotic resistance and the upcoming next-generation PCVs highlight the need to continue to monitor the national dynamic of pneumococcal nasopharyngeal carriage.

## Figures and Tables

**Figure 1 antibiotics-12-01020-f001:**
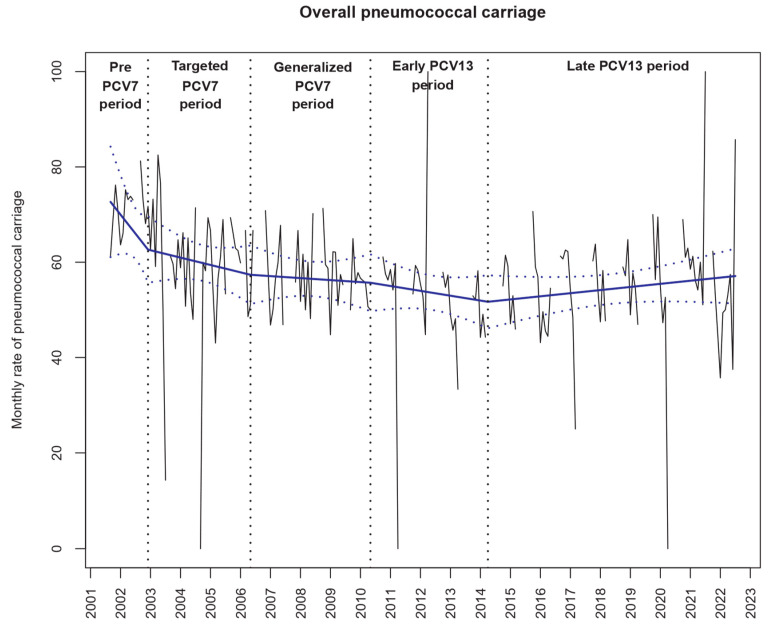
Evolution of overall *Streptococcus pneumoniae* (Sp) carriage in children with acute otitis media enrolled during the study period (*n* = 17,136). Notes: The black lines show the observed data. Using a segmented regression model, we estimated the rates over time (blue slope lines) and its 95% confidence interval (blue dotted lines). The “pre-PCV7 period” was from September 2001 to December 2002, the “targeted PCV7 period” from January 2003 to May 2006, the “PCV7 period” from June 2006 to May 2010, the “early PCV13 period” from June 2010 to April 2014 and the “late PCV13 period” from May 2014 to July 2022.

**Figure 2 antibiotics-12-01020-f002:**
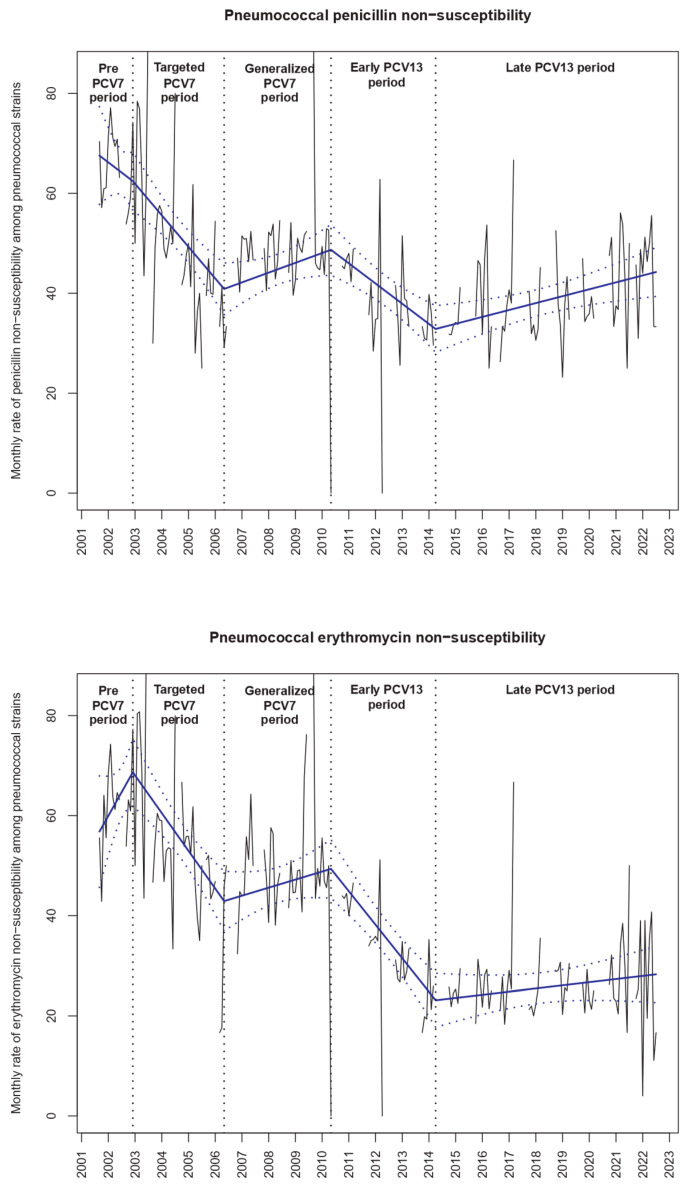
Evolution of penicillin and erythromycin non-susceptibility among the pneumococcal strains isolated from nasopharyngeal carriage. Notes: The black lines show the observed data. Using a segmented regression model, we estimated the rates over time (blue slope lines) and its 95% confidence interval (blue dotted lines). The “pre-PCV7 period” was from September 2001 to December 2002, the “targeted PCV7 period” from January 2003 to May 2006, the “PCV7 period” from June 2006 to May 2010, the “early PCV13 period” from June 2010 to April 2014 and the “late PCV13 period” from May 2014 to July 2022.

**Figure 3 antibiotics-12-01020-f003:**
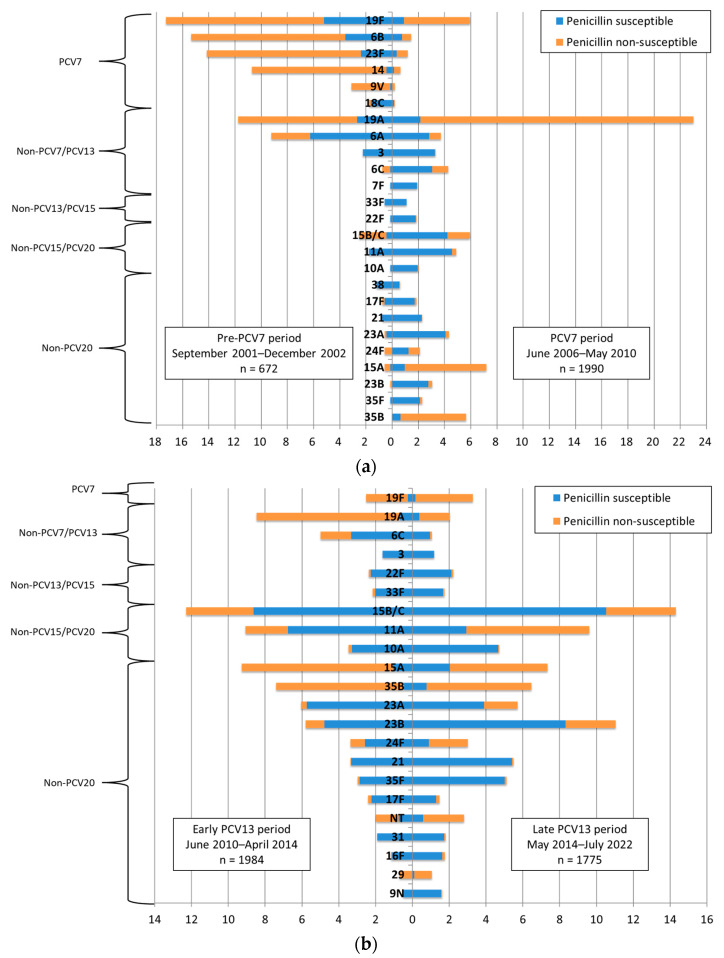
(**a**,**b**). Serotype distribution during the pre-PCV7, PCV7, early PCV13 and late PCV13 periods. The “pre-PCV7 period” was from September 2001 to December 2002, the “PCV7 period” from June 2006 to May 2010, the “early PCV13 period” from June 2010 to April 2014 and the “late PCV13 period” from May 2014 to July 2022. Note: Serotypes accounting for <1% in both periods are not shown. PCV7, 7-valent pneumococcal conjugate vaccine; PCV13, 13-valent pneumococcal conjugate vaccine; NT, non-typable.

**Figure 4 antibiotics-12-01020-f004:**
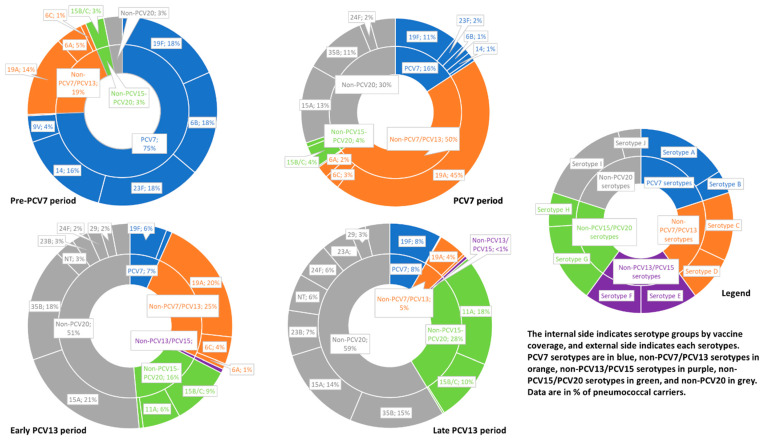
Serotype distribution among penicillin non-susceptible Sp strains isolated during the pre-PCV7, PCV7, early PCV13 and late PCV13 periods. Notes: The “pre-PCV7 period” was from September 2001 to December 2002, the “PCV7 period” from June 2006 to May 2010, the “early PCV13 period” from June 2010 to April 2014 and the “late PCV13 period” from May 2014 to July 2022. The “early PCV13 period” was from June 2010 to April 2014 and the “late PCV13 period” from May 2014 to July 2022. Serotypes isolated in <1.0% of patients during each period are not shown.

**Table 1 antibiotics-12-01020-t001:** Characteristics of the 17,136 children with acute otitis media by study period.

Periods	Pre-PCV7 *n* = 943	Targeted PCV7 *n* = 2314	PCV7 *n* = 3458	Early PCV13 *n* = 3677	Late PCV13 *n* = 6744	Total *n* = 17,136
Male, *n* (%)	524 (55.6)	1225 (52.9)	1833 (53.0)	1972 (53.6)	3605 (53.4)	9159 (53.4)
Age (months), median (IQR)	13.1 (9.4–17.6)	12.9 (9.3–17.5)	12.7 (9.4–17.2)	13.0 (9.5–17.5)	12.6 (9.2–17.2)	12.8 (9.3–17.3)
Daycare attendance, *n* (%)	316 (33.5)	764 (33.0)	1416/3455 (41.0)	1769 (48.1)	3982/6741 (59.1)	8247/17,130 (48.1)
Siblings, *n* (%)	510 (54.1)	1369 (59.2)	1949/3457 (56.4)	2057 (55.9)	3878/6743 (57.5)	9763/17,134 (9763)
Otalgia, *n* (%)	811 (86.0)	1954 (84.4)	2565/3455 (74.2)	2668 (72.6)	4444/6727 (66.1)	12,442/17,116 (72.7)
Fever (≥38.5 °C), *n* (%)	775/775 (100.0)	1968/1986 (99.1)	2047/3429 (59.7)	2041/3646 (56.0)	3538/6646 (53.2)	10,369/16,482 (62.9)
Conjunctivitis, *n* (%)	NA	NA	872/3447 (25.3)	1048 (28.5)	1847/6740 (27.4)	3767/13,864 (27.2)
Otorrhea, *n* (%)	101 (10.7)	240 (10.4)	293/3458 (8.5)	256 (7.0)	498/6741 (7.4)	1388/17,133 (8.1)
Bilateral AOM, *n* (%)	NA	NA	482/886 (54.4)	1844 (50.1)	3063/6738 (45.5)	5389/11,301 (47.7)
History of AOM, *n* (%)	NA	NA	983/1755 (56.0)	2209 (55.2)	3363 (49.9)	6375/12,176 (52.4)
Otitis prone children, *n* (%)	NA	NA	342/1755 (19.5)	644/3677 (17.5)	985/6743 (14.6)	1971/12,175 (16.2)
Recent use of antibiotics, *n* (%)	448 (47.2)	964 (41.7)	1634/3455 (47.3)	1575/3676 (42.8)	2451/7743 (36.3)	7069/17,131 (41.3)
Broad-spectrum antibiotics, *n* (%)	375/442 (84.8)	580/959 (88.6)	1512/1629 (92.8)	974/1571 (62.0)	793/2433 (32.6)	4504/7034 (64.0)

The “pre-PCV7 period” was from September 2001 to December 2002, the “targeted PCV7 period” from January 2003 to May 2006, the “PCV7 period” from June 2006 to May 2010, the “early PCV13 period” from June 2010 to April 2014 and the “late PCV13 period” from May 2014 to July 2022. In the pre PCV7 and the targeted PCV7 periods, only children with fever and/or otalgia were enrolled in the study. From the PCV7 period, all children with an AOM, whatever the associated signs, were enrolled. Data on bilateral AOM, history of AOM and otitis-prone children were available from only October 2008. For clarity, denominators are only shown if data are missing. IQR, interquartile range; AOM, acute otitis media; PCV, pneumococcal conjugate vaccine.

**Table 2 antibiotics-12-01020-t002:** Carriage, serotype distribution and resistance of pneumococcus strains isolated in children with acute otitis media by study period.

Periods	Pre-PCV7 *n* = 943	Targeted PCV7 *n* = 2314	PCV7 *n* = 3458	Early PCV13 *n* = 3677	Late PCV13 *n* = 6744	Total *n* = 17,136
Pneumococcal carriage	672 (71.3)	1418 (61.3)	1990 (57.5)	1984 (54.0)	3775 (56.0)	9839 (57.4)
Monthly trend (95% CI)	−0.49% (−1.22 to +0.24), *p* = 0.18	−0.11% (−0.34 to +0.12), *p* = 0.35	−0.03% (−0.17 to +0.11), *p* = 0.66	−0.08% (−0.64 to +0.48), *p* = 0.78	+0.05% (−0.04 to +0.14), *p* = 0.29	
PCV7 serotypes	421 (44.6)	565 (24.4)	193 (5.6)	64 (1.7)	132 (2.0)	1375 (8.0)
PCV13 + 6C serotypes	585 (62.0)	984 (42.5)	925 (26.7)	387 (10.5)	306 (4.5)	3187 (18.6)
6 additional serotypes + serotype 6C	164 (17.4)	419 (18.1)	732 (21.2)	323 (8.8)	174 (2.6)	1812 (10.6)
PCV15 serotypes	590 (87.8)	1036 (73.1)	985 (49.5)	477 (24.0)	456 (12.1)	3544 (36.0)
PCV20 serotypes	620 (92.3)	1160 (81.8)	1247 (62.7)	986 (49.7)	1565 (41.5)	5578 (56.7)
Penicillin non-susceptible strains	442/671 (65.9)	687/1414 (48.6)	944/1987 (47.5)	774/1983 (39.0)	1439/3776 (38.1)	4286/9831 (43.6)
Monthly trend (95% CI)	−0.24% (−0.85 to +0.36), *p* = 0.41	−0.31% (−1.92 to +1.31), *p* = 0.70	+0.18% (+0.09 to +0.27), *p* < 0.001	−0.41% (−0.66 to −0.16), *p* = 0.001	+0.15% (+0.08 to 0.22), *p* < 0.001	
Penicillin resistant strains	8/671 (1.2)	7/1414 (0.5)	3/1987 (0.1)	10/1983 (0.5)	18/2337 (0.5)	46/9831 (0.5)
Erythromycin non-susceptible strains	423/672 (62.9)	749/1415 (52.9)	969/1987 (48.8)	663/1984 (33.4)	956/3776 (25.3)	3760/9834 (38.2)
Monthly trend (95% CI)	+0.63% (−0.12 to +1.39), *p* = 0.09	−0.56% (−1.00 to −0.12), *p* = 0.01	+0.14% (+0.07 to +0.22), *p* < 0.001	−0.78% (−1.16 to −0.39), *p* < 0.001	+0.10% (+0.06 to +0.14), *p* < 0.001	
Most frequent serotypes	19F, 116 (17.3)	19F, 235 (16.6)	19A, 458 (23.0)	15B/C, 244 (12.3)	15B/C, 540 (14.3)	15B/C, 978 (9.9)
	6B, 103 (15.3)	19A, 190 (13.4)	15A, 143 (7.2)	15A, 184 (9.3)	23B, 417 (11.0)	19A, 973 (9.9)
	23F, 95 (14.1)	23F, 109 (7.7)	15B/C, 118 (5.9)	11A, 180 (9.1)	11A, 363 (9.6)	11A, 697 (7.1)
	19A, 79 (11.8)	6B, 108 (7.6)	19F, 118 (5.9)	19A, 169 (8.5)	15A, 278 (7.4)	19F, 643 (6.5)
	14, 72 (10.7)	3, 90 (6.3)	35B, 112 (5.6)	35B, 147 (7.4)	35B, 244 (6.5)	15A, 639 (6.5)
Most frequent penicillin non-susceptible serotypes	19F, 81 (18.3)	19A, 172 (25.0)	19A, 414 (43.9)	15A, 163 (21.1)	11A, 252 (17.5)	19A, 865 (20.2)
	23F, 79 (17.9)	19F, 162 (23.6)	15A, 124 (13.1)	19A, 155 (20.0)	35B, 215 (14.9)	15A, 515 (12.0)
	6B, 79 (17.9)	23F, 92 (13.4)	19F, 120 (10.6)	35B, 138 (17.8)	15A, 200 (13.9)	19F, 506 (11.8)
	14, 69 (15.6)	6B, 68 (9.9)	35B, 99 (10.5)	15B/C, 73 (9.4)	15B/C, 143 (9.9)	35B, 462 (10.8)
	19A, 61 (13.8)	14, 51 (7.4)	15B/C, 34 (3.6)	11A, 46 (5.9)	19F, 118 (8.2)	11A, 305 (7.1)

Notes: Data are *n* (%); denominators are only shown if data are missing. The “pre-PCV7 period” was from September 2001 to December 2002, the “targeted PCV7 period” from January 2003 to May 2006, the “PCV7 period” from June 2006 to May 2010, the “early PCV13 period” from June 2010 to April 2014 and the “late PCV13 period” from May 2014 to July 2022. Monthly trends are estimated with a segmented regression model. PCV7 (4, 6B, 9V, 14, 18C, 19F and 23F), PCV13 (PCV10 + 3, 6A and 19A), PCV15 (PCV13 + 22F and 33F), PCV20 (PCV15 + 8, 10A, 11A, 12F and 15B). PCV, pneumococcal conjugate vaccine.

## Data Availability

Data are available upon reasonable request to the corresponding author.

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
