# Peer review of "Dynamics of Antibiotic Resistance of Streptococcus pneumoniae in France: A Pediatric Prospective Nasopharyngeal Carriage Study from 2001 to 2022"

_antibiotics, 2023, doi:10.3390/antibiotics12061020_

Round 1
Reviewer 1 Report
General Comments:
1. After a critical review of this 21-year extended prospective study, it was possible to understand that the authors (Rybak, et al.) intend to identify trends related to nasopharyngeal colonization, serotype distribution and antibiotic resistance of Streptococcus pneumoniae among children with Acute Otitis Media in France.
2. The comparability and reproducibility of this study are assured, since the manuscript describes very well the experimental procedures and methodologies used to collect the data and to perform the statistical analysis.
3. The study design is appropriate to address the research question and provides comprehensive insights on the dynamics of pneumococcal nasopharyngeal colonization, antibiotic resistance, serotype distribution and serotype replacement, improving our understanding of the complex scenario of the pneumococcal epidemiology in France.
4. The cited reference list appears to be relevant to the topic, providing additional information for readers who want to learn more details. Twenty-three of 35 references are from the last 5 years, 7/35 are from 6 to 10 years ago and only 4 of 96 are older than 10 years.
5. However, the authors provide a poor introduction with insufficient background on the topics of the manuscript. A better overview of the relevant topics would improve readers understanding, especially those who are not specialist on Streptococcus pneumoniae and would help to strengthen the discussion based on the results.
6. The results presented in the manuscript, figures and tables (main text and supplements) are well-organized and clearly presented, making them easy to understand for readers.
7. The conclusions drawn by the authors are supported by the results of this research. However, they should not be generalized beyond France and beyond the limitations the authors discuss for their study design. Pneumococcal vaccination programs in many countries are different from that of France. Hence, serotype distribution and serotype replacement depends on the geographic region we investigate. Even in France, nasopharyngeal carriage dynamics can be altered by many other risk factors, beyond AOM. Antibiotic resistance also depends on the clonality of the pneumococcal strains and isolates. So, discussion, conclusions and the title of this manuscript should be kept in the limits of the study design and should be contextualize to nasopharyngeal carriage among children with AOM in France.
Specific Comments:
8. Authors should provide a short sentence (with references) explaining to readers why AOM increase the probability of nasopharyngeal carriage of Streptococcus pneumoniae.
9. If available, authors should consider to include any MLST data and analysis to strengthen their findings and explanations related to antibiotic resistance. The explanation to the findings are not only in the serotype distribution.
Please, correct some minor spelling errors throughout the text:
Streptococcus pneumoniae in italics (References).
Sp in biology also means "Species". Better use S. pneumoniae.
It is suggested to avoid the word "unique" in the line 184
Author Response
General Comments:
After a critical review of this 21-year extended prospective study, it was possible to understand that the authors (Rybak, et al.) intend to identify trends related to nasopharyngeal colonization, serotype distribution and antibiotic resistance of Streptococcus pneumoniae among children with Acute Otitis Media in France.
The comparability and reproducibility of this study are assured, since the manuscript describes very well the experimental procedures and methodologies used to collect the data and to perform the statistical analysis.
The study design is appropriate to address the research question and provides comprehensive insights on the dynamics of pneumococcal nasopharyngeal colonization, antibiotic resistance, serotype distribution and serotype replacement, improving our understanding of the complex scenario of the pneumococcal epidemiology in France.
The cited reference list appears to be relevant to the topic, providing additional information for readers who want to learn more details. Twenty-three of 35 references are from the last 5 years, 7/35 are from 6 to 10 years ago and only 4 of 96 are older than 10 years.
We thank the reviewer for these comments.
However, the authors provide a poor introduction with insufficient background on the topics of the manuscript. A better overview of the relevant topics would improve readers understanding, especially those who are not specialist on Streptococcus pneumoniae and would help to strengthen the discussion based on the results.
As requested by the reviewer, we have deeply modified the first paragraph of the introduction to help non-specialized readers to understand the importance of S. pneumoniae nasopharyngeal carriage surveillance (lines 44 to 60).
“The nasopharynx, particularly in children, is the ecological niche of pneumococci and the site of selective pressure due to antibiotics exposure and conjugate vaccines. Nasopharyngeal (NP) carriage is a prerequisite for pneumococcal disease. Infections due to S. pneumoniae include acute otitis media (AOM), which is in many countries the main cause for antibiotic prescription in children. Pneumococcus is also implicated in less-frequent other respiratory tract infections such as sinusitis and pneumonia, and invasive pneumococcal infections. Monitoring S. pneumoniae strains involved in non-invasive infections is a challenge. Indeed, tympanocentesis is no longer performed in most countries for first-line treatment of AOM. Therefore, carriage surveillance has been proposed as a proxy to monitor the serotype distribution and antibiotics resistance of pneumococci for AOM, especially after pneumococcal conjugate vaccine implementation. In many countries, antibiotics are recommended for AOM in young children. Pneumococcus and Haemophilus influenzae resistance to antibiotics are among the main determinants for the choice and the dosage of first-line antibiotic. Furthermore, NP carriage studies are important to try to understand the complex interactions of PCV implementation and antibiotics use in S. pneumoniae carriage and infections.”
The results presented in the manuscript, figures and tables (main text and supplements) are well-organized and clearly presented, making them easy to understand for readers.
We thank the reviewer for this remark.
The conclusions drawn by the authors are supported by the results of this research. However, they should not be generalized beyond France and beyond the limitations the authors discuss for their study design. Pneumococcal vaccination programs in many countries are different from that of France. Hence, serotype distribution and serotype replacement depends on the geographic region we investigate. Even in France, nasopharyngeal carriage dynamics can be altered by many other risk factors, beyond AOM. Antibiotic resistance also depends on the clonality of the pneumococcal strains and isolates. So, discussion, conclusions and the title of this manuscript should be kept in the limits of the study design and should be contextualize to nasopharyngeal carriage among children with AOM in France.
We thank the reviewer for this remark.
We have changed the title to: “Dynamics of Antibiotic Resistance of Streptococcus pneumoniae in France: a Pediatric Prospective Nasopharyngeal Carriage Study from 2001 to 2022”.
Furthermore, we have completed the discussion section as follows (lines 261 to 266): “As discussed previously, France has high PCV13 vaccine coverage, major community antibiotics consumption and a high proportion of children attending daycare. Furthermore, clonal changes in the pneumococcal population in carriage may differ from country to country. This combination of factors may prevent the generalization of our results.”
We also have changed the first and last sentence of the conclusion (lines 275 to 276): “After the implementation of PCV7 and PCV13, pneumococcal antibiotic resistance significantly decreased in France.” and lines 280 “Expected changes encompassing replacement together with uprising antibiotic resistance and the upcoming next-generation PCVs highlight the need to continue to monitor the national dynamic of pneumococcal nasopharyngeal carriage.”
Specific Comments:
Authors should provide a short sentence (with references) explaining to readers why AOM increase the probability of nasopharyngeal carriage of Streptococcus pneumoniae.
We thank the reviewer for this remark. We have added the following sentences (lines 89 to 92) with references: “Previous studies have shown than children with AOM have higher rates of pneumococcal carriage compared to healthy children. Enrollment of these children allowed us to monitor more easily serotype distribution and antibiotic susceptibility over time by reducing number of children swabbed.” with the following references: Faden et al, Relationship between nasopharyngeal colonization and the development of otitis media in children, J Infect Dis. 1997 and Revai et al, Association of nasopharyngeal bacterial colonization during upper respiratory tract infection and the development of acute otitis media, Clin Infect Dis.
- If available, authors should consider to include any MLST data and analysis to strengthen their findings and explanations related to antibiotic resistance. The explanation to the findings are not only in the serotype distribution.
We agree with the reviewer. However, MLST data are not available.
We have modified the discussion section to further discuss this limitation (lines 263 to 268): “Furthermore, clonal changes in the pneumococcal population in carriage may differ from country to country. This combination of factors may prevent the generalization of our results. Finally, whole-genome sequencing and multilocus sequencing typing (MLST) are not routinely performed in our study. These techniques could strengthen the relation between serotype and antibiotic resistance.”
Comments on the Quality of English Language
Please, correct some minor spelling errors throughout the text:
Streptococcus pneumoniae in italics (References).
Sp in biology also means "Species". Better use S. pneumoniae.
We thank the reviewer for this remark. We have changed this term throughout the manuscript. Furthermore, we have defined the abbreviation PNSP (penicillin non-susceptible S. pneumoniae).
It is suggested to avoid the word "unique" in the line 184
We have deleted this term.
Reviewer 2 Report
In the manuscript Rybak et al., analyze the carriage, serotypes, and penicillin and erythromycin resistance of S. pneumoniae in the nasopharynx of children from 6 months to 2 years of age during the period 2001-2022. During this period, the PCV7 and PCV13 vaccines were implemented, so it is interesting to monitor the pneumococcal nasopharyngeal carriage. The aim of the study was also to establish if the carriage of serotypes resistant, PCV serotypes is associated with any of the factors studied such as age, previous acute otitis media, etc.
I read your article with great interest and would like to present forward the following comments/ suggestions:
lane 79. Briefly describe the methodology used, described in the referenced articles.
Supplemental Figure 1. Expand the figure caption with a brief explanation of each graph, meaning of the continuous lines, dashed lines, colors, arrows, etc.
lane 133. Hi carriage was analyzed; however, it is not mentioned in Results.
lane 199. The sentence needs a reference or data that supports the statement.
Quality of English Language
Author Response
In the manuscript Rybak et al., analyze the carriage, serotypes, and penicillin and erythromycin resistance of S. pneumoniae in the nasopharynx of children from 6 months to 2 years of age during the period 2001-2022. During this period, the PCV7 and PCV13 vaccines were implemented, so it is interesting to monitor the pneumococcal nasopharyngeal carriage. The aim of the study was also to establish if the carriage of serotypes resistant, PCV serotypes is associated with any of the factors studied such as age, previous acute otitis media, etc.
I read your article with great interest and would like to present forward the following comments/ suggestions:
lane 79. Briefly describe the methodology used, described in the referenced articles.
We thank the reviewer for this remark.
We have added all the details in the methods section as follows (lines 81 to 97): “Between September 2001 and July 2022, 161 pediatricians throughout France participated to this prospective population-based surveillance study. Pediatricians are part of a research and teaching network (ACTIV). The methodology was previously described. From September to July of each subsequent year, we enrolled children from both sex with AOM who were 6 to 24 months old. Diagnostic criteria for AOM included the algorithm proposed by Paradise for acute suppurative otitis media (effusion plus marked redness, marked bulging or moderate redness and bulging) associated with fever and/or otalgia and/or irritability. Previous studies have shown than children with AOM have higher rates of pneumococcal carriage compared to healthy children. Enrollment of these children allowed us to monitor more easily serotype distribution and antibiotic susceptibility over time by reducing number of children swabbed. We excluded children with antibiotics treatment within 7 days before enrolment, severe underlying disease, or inclusion in the study during the previous 12 months. The physician collected medical history, antibiotic consumption during the 3 months before inclusion and type of antibiotics used, immunization history, daycare attendance and clinical examination findings on an electronic case report form.”
We have also completed lines 103 to 108: “The techniques of NP swabbing and microbiologic methods were previously de-scribed, and remained unchanged over the whole study period. Specimens were placed in transport medium and transferred within 48 hours for analyzised at the National Reference Center for Pneumococci (NRCP; Centre Hospitalier Intercommunal de Créteil, Créteil, France) or at Robert Debré Hospital (Paris, France). S. pneumoniae strains were identified with morphology and standard methods.”
Supplemental Figure 1. Expand the figure caption with a brief explanation of each graph, meaning of the continuous lines, dashed lines, colors, arrows, etc.
We thank the reviewer for this remark.
We have changed the figure legend: from “The blue slope lines were estimated using a segmented regression model. The blue dotted lines show the 95% confidence interval estimated using the segmented regression model.” to “Using a segmented regression model, we have estimated the rates over time (blue slope lines) and its 95% confidence interval (blue dotted lines).” and slightly reduced the size of the Supplemental Figures. Thanks to these modifications, figure and its legend are on the same page facilitating the readers interpretation.
lane 133. Hi carriage was analyzed; however, it is not mentioned in Results.
We thank the reviewer for noticing this error. In this specific article, the analysis concerned only S. pneumoniae.
lane 199. The sentence needs a reference or data that supports the statement.
We thank the reviewer for this remark. We have added an article supporting this statement (https://doi.org/10.1086/428585) in addition to the one already cited (https://doi.org/10.2165/00148581-200810020-00002).
Reviewer 3 Report
Rybak et al show a very interesting study evaluating the serotype distribution and antibiotic resistance of pneumococcal strains associated to nasopharyngeal carriage in the last 21 years.
The manuscript is well written but there are some points that need to be addressed:
1. PCV13 periods: The early PCV13 period contains up to 4 years (2010 to 2014) whereas the late PCV13 period contains up to 8 years (2014 to 2022). This variation affect some of the results that it seems that the impact of PCV13 is lower in terms of number of circulating strains in carriage. Inclusion of a period analyzing the middle effect of PCV13 would be interesting and it would reduce the differences of table 2
2. Figure 3 it is very difficult to interpret. I would suggest changing this figure for a table or indicating with more clarity the periods within the figure.
3. Figure 4 is also difficult to interpret. The figure legend should indicate that in each circle the external side indicate serotypes whereas the internal side indicate serotype groups by vaccine coverage. I also suggest to change the colors between external and internal side to clarify the reader
4. I suggest to cite in the discussion section two recent manuscripts showing the impact of PCV7, PCV13 and COVID-19 in antibiotic resistance and serotypes evolution of S. pneumoniae (Sempere et al, Lancet Microbe. 2022 Oct;3(10):e744-e752. doi: 10.1016/S2666-5247(22)00127-6; Sempere et al J Antimicrob Chemother. 2022 Mar 31;77(4):1045-1051. doi: 10.1093/jac/dkab482). Some of the results of the study by Rybak et al correlates very well with recent findings described in the studies suggested above such as the increase of non-susceptible strains to penicillin and erythromycin in the late PCV7 period (mainly due to serotype 19A) and the increase in the late PCV13 period due to serotype 11A. The impact of COVID-19 increasing antibiotic resistance in the first pandemic year due to generic consumption of antibiotics is interesting to mention in the discussion section of Rybak`s manuscript. This is especially important for serotype 11A as in Sempere et al, the MIC90 for this serotype increased from 2 µg/ml in the pre-pandemic period to 4 µg/ml in the pandemic period which is considered resistant by EUCAST whereas in the suppl fig 4 by Rybak et al, they show a permanent resistant increase by this serotype which is consistent with the study by Sempere et al. Overall, the study by Rybak et al is of great importance because carriage and IPD are generally well correlated and similar trends of antibiotic resistance in carriage are found in IPD.
